

# Testing relationship between plant productivity and diversity in a desertified steppe in Northwest China

Yang Yang[1,2,3] and Bingru Liu[1]

[1] Key Lab of Restoration and Reconstruction of Degraded Ecosystem in Northwest China of Ministry of Education, Ningxia University, Yinchuan, China
[2] State Key Laboratory of Loess and Quaternary Geology, Institute of Earth Environment, Chinese Academy of Sciences, Xi'an, China
[3] CAS Center for Excellence in Quaternary Science and Global Change, Xi'an, China

Corresponding author
Bingru Liu, bingru.liu@163.com

## ABSTRACT

The rapid global plant diversity and productivity loss has resulted in ecosystem functional degeneration in recent decades, and the relationship between plant diversity and productivity is a pressing issue around the world. Here, we sampled six plant communities that have not been grazed for 20 years, i.e., *Agropyron mongolicum*, *Stipa bungeana*, *Cynanchum komarovii*, *Glycyrrhiza uralensis*, *Sophora alopecuroides*, *Artemisia ordosica*, located in a desertified steppe, northwestern China, and tested the relationship between plant diversity and productivity in this region. We found a positive linear relationship between AGB (above-ground biomass) and BGB (below-ground biomass), and the curves between plant diversity and AGB were unimodal ($R^2 = 0.4572$, $p < 0.05$), indicating that plant productivity increased at a low level of diversity but decreased at a high level of diversity. However, there was no significant relationship between BGB and plant diversity ($p > 0.05$). Further, RDA (redundancy analysis) indicated that soil factors had a strong effect on plant diversity and productivity. Totally, GAMs (generalized additive models) showed that soil factors (especially total nitrogen TN, total carbon TC, soil microbial biomass nitrogen SMB-N, soil microbial biomass carbon SMB-C) explained more variation in plant diversity and productivity (78.24%), which can be regarded as the key factors driving plant diversity and productivity. Therefore, strategies aiming to increase plant productivity and protect plant diversity may concentrate on promoting soil factors (e.g., increasing TC, TN, SMB-N and SMB-C) and plant species, which can be regarded as an effective and simple strategy to stabilize ecosystems to mitigate aridity in desertified steppes in northwestern China.

## INTRODUCTION

Grasslands are widely distributed around the world, which cover about a quarter of the surface of the Earth, playing a crucial role in the global C (carbon) cycling (*Loreau, Naeem & Inchausti, 2001*; *Cardinale, Srivastava & Duffy, 2006*). Most studies on grasslands have emphasized the importance of plant diversity, which can be regarded as an essential part of terrestrial ecosystems (*Hooper, Chapin Iii & Ewel, 2005*; *Hooper, Chapin Iii & Ewel, 2005*;

*Hector & Bagchi, 2007*). Plant productivity provide a amount of available energy due to complementary resource utilization in ecosystems. In theory, the relationship between plant diversity and productivity presented in positive in most grasslands around the world (*Bai, Wu & Clark, 2012*; *Tang et al., 2018*). Actually, the higher plant productivity is expected to increase plant diversity because of the abundant food resources (*Baldock & Sibly, 1990*; *Worm, Barbier & Beaumont, 2006*); in turn, the higher plant diversity can also promote an increase in productivity due to the higher multiplicity and quantity of plant species (*Tilman & Downing, 1994*; *Ives & Carpenter, 2007*; *Chen et al., 2018*). Thus, two issues should be addressed: (1) Is plant productivity controlled by diversity in grasslands around the world? (2) Is there truly a general relationship between plant diversity and productivity around the world? Recently, hump-shaped relationships have been regarded as the most widespread pattern between plant diversity and productivity, showing that plant diversity get peaks at an intermediate plant productivity level (*Tilman, Wedln & Knops, 1996*; *Hooper, Bignell & Brown, 2000*; *Loreau, Naeem & Inchausti, 2001*). For example, *Mittelbach, Steiner & Scheiner (2001)* reported a hump-shaped curve between plant productivity and diversity at the local scale based on a meta-analysis, whereas *Whittaker & Heegaard (2003)* noted several problems in their statistical methods (meta-analysis). To solve this issue, *Gillman & Wright (2006)* re-analyzed the relationship between plant diversity and productivity, and then found a monotonic curve at the global scale and a hump-shaped curve at the regional scale. On the contrary, *Bai, Wu & Clark (2012)* reported a linear relationship in grasslands of northern China. To sum, the relationship between plant diversity and productivity in grasslands are still controversial until now (*Declerck, Vandekerkhove & Johansson, 2005*; *Thiele-Bruhn & Beck, 2005*). Due to human disturbance, niche specialization, and spatial scales in different ecosystems, the relationship between plant diversity and productivity may be negative (*Tilman, Knops & Wedin, 1997*; *Gillman & Wright, 2006*), positive (*Wang, Long & Ding, 2001*; *Yang, Rao & Hu, 2003*), flat (non-significant) (*Hooper, Bignell & Brown, 2000*), U-shaped (*Wang, Long & Ding, 2001*; *Bai, Wu & Clark, 2012*), hump-shaped (*Mittelbach, Steiner & Scheiner, 2001*; *Tilman, Reich & Isbell, 2012*; *Backes & Veen, 2013*) for all kinds of grasslands around the world, and these multiple patterns between plant diversity and productivity exist due to different explanations, including the availability of resources and energy, disturbance, plant species pool, herbivory, the spatial heterogeneity, history of plant communities (*Tilman & Downing, 1994*; *Cardinale, Wright & Cadotte, 2007*; *Sun, Cheng & Li, 2013*).

Recently, numerous published studies on the prevailing view of the relationship between plant diversity and productivity in grasslands have been based on either meta-analysis or theoretical arguments (*Trax, Brunow & Suedekum, 2015*; *Cardinale, Duffy & Gonzalez, 2012*; *Craven, Isbell & Manning, 2016*), and these studies have been conducted mainly in Africa, Europe, China, and North America. However, there are most problems related to the data, including plant sample sizes and the methods for plant productivity (*Bai, Wu & Clark, 2012*; *Cardinale, Duffy & Gonzalez, 2012*). For example, Grace suggested that plant productivity is only one of the most influential factors affecting plant diversity (*Grace, 1999*). Other research processes, such as spatial heterogeneity, human disturbance, may also affect plant diversity. On the basis of considerable empirical and theoretical evidence

and data, *Abrams (1995)* noted that the processes of coexistence and competition might lead to a monotonic relationship between productivity and diversity. In Tibetan Plateau's alpine grasslands, Sun et al. reported the environmental factors that affected above-ground biomass based on a meta-analysis, and they suggested that the patterns of plant biomass cannot be explained by a single environmental factor (*Sun, Cheng & Li, 2013*; *Sun & Wang, 2016*). Further, *Sun, Ma & Lu (2018)* explored the trade-offs between BGB and AGB, and BGB and AGB were strongly affected by soil nutrients. Additionally, *Porazinska et al. (2018)* reported that above-ground diversity was strong related to below-ground diversity in an alpine ecosystem. *Yang et al. (2018)* studied how biotic and abiotic factors modulate plant biomass in grasslands on the Loess Plateau. Although these published studies are scientific and useful, the relationship between plant productivity and diversity in the desertified steppe of China is still largely unknown. To our knowledge, the desertified steppe of China is well known by the extreme drought, and vegetation in this region is slow to recover. Most seriously, plant productivity is also lower than the other grasslands because of human disturbances and climate conditions, leading to much attention in terms of the stability of ecological restoration in northwestern China (*Liu, Zhao & Zhao, 2012*; *Yang, Liu & An, 2018*). Thus, it needed to determine the driving factors for plant diversity and productivity in this region.

Here, we attempted to explain the relationship and the main driving factors of plant diversity and productivity in the desertified steppe, northwestern China. Thereafter, we addressed three specific issues: (1) Compared to the grassland at the level of China or the world, how does the plant diversity and productivity change in the desertified steppe? (2) What patterns of the relationship between plant diversity and productivity present? (3) What are the dominant driving factors for plant diversity and productivity in this region? These specific issues are crucial for the biodiversity conservation and ecological restoration in the desertified steppe in northwestern China.

## MATERIALS AND METHODS

### Study site

This research started in the desertified steppe of northwestern China, Ningxia Hui Autonomous Region of China (106°30′–107°41′E, 37°04′–38°10′N), and was approval by The Grassland Committee of Ningxia Hui Autonomous Region (#15032). The study site had a continental, temperate, monsoon, semiarid climate. The mean annual potential evaporation is approximately 1223.8–2087.6 mm, and the average elevation is 1450 m. Further, the mean annual precipitation is approximately 298 mm over the past 12 years. In July, it has the highest monthly mean temperature (22.4 °C), and in January it has the lowest monthly mean temperature (8.7 °C), respectively. The soil type is mainly dominant by the desertification sierozem with the lower soil fertility, and plants are solitary and show drought-resistant characteristics, including some small xerophytic shrubs and annual weeds. The leaves of plants generally have clear xerophytic morphological characteristics, mainly including members of *Asteraceae, Gramineae, Zygophyllaceae, Liliaceae, Cruciferae* and *Leguminosae* (*Li, 2001*; *Pei, Fu & Wan, 2008*; *Liu et al., 2011*).
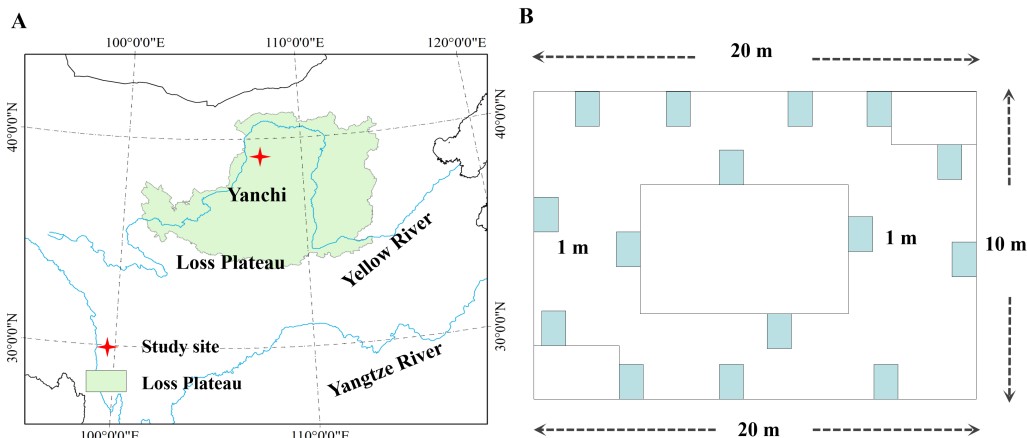

**Figure 1** **Location of the study area in desertified steppe.** (A) Study site; (B) the sampling methods. Chinese map was created by ArcGIS 9.3 software, http://www.arcgis.com/features/; data was collected from Google Earth ©2017 satellite imagery.

## Sampling methods

Six plant communities were sampled, which have not been grazed for 20 years, i.e., *Agropyron mongolicum*, *Stipa bungeana*, *Cynanchum komarovii*, *Glycyrrhiza uralensis*, *Sophora alopecuroides*, *Artemisia ordosica*, which are dominant plant species in the desertified steppe in northwestern China. AGB and BGB were sampled at their peak times in July and August in 2015. Each plant community consisted of five sampling sites (100 ×100 m), and thirty sampling sites were selected from within the six plant communities (Fig. 1). In order to make this study representative, each sampling site was spaced approximately 1 km apart with the same natural conditions. Data were collected by previous study in *Yang et al. (2018)*. Specially, sampled 15 quadrats ($1 \times 1$ m$^2$) for herbaceous plants and 15 quadrats ($3 \times 3$ m$^2$) for shrubs around with 200 m$^2$, and AGB and BGB of all plant species in each quadrat was harvested. Meanwhile, we measured plant height (the maximum, plant species abundance, plant cover, and then dug up the whole root (measured BGB) in each quadrat. The collected AGB and BGB were dried (dry mass) at 85 °C for more than 72 h until a constant weight, and then weighed by electronic scales. In addition, the altitude, latitude, longitude were recorded by a GPS receiver. We investigated the growth characteristics of these plant species and morphological characteristics, which partly reflected the general characteristics of plant communities.

Each soil sample was collected in triplicate from 0 cm to 15 cm of soil profile, and separated into two parts: (1) One part of the fresh soil sample was oven-dried by aluminium container to measure SW (soil water content, %); (2) The other part of the fresh soil sample was used to measure soil physicochemical properties and nutrients by passing through a 2 mm sieve.

## Soil sample analysis

Soil pH and EC (soil electrical conductivity, $\mu$scm$^{-2}$) were measured in 1:1 (v/v) and 1:5 (v/v) soil water solution, respectively. BD (soil bulk density, gcm$^{-3}$) was measured by

oven-dried (for 72 h until the constant weight). TC (soil total carbon, $gkg^{-1}$) and TN (soil total nitrogen, $gkg^{-1}$) were measured by an elemental analyzer (elementar vario MACRO cube, Germany). AN (soil available nitrogen, $mgkg^{-1}$) was measured by $NaOH-H_3BO_3$. AP (available phosphorus, $mgkg^{-1}$) and TP (total phosphorus, $gkg^{-1}$) were measured by molybdenum-antimony using model 722-Spectrometer. Finally, we used fumigation-extraction method to measure and calculate SMB-C (soil microbial biomass C, $mgkg^{-1}$) and SMB-N (soil microbial biomass N, $mgkg^{-1}$) (*Liu et al., 2011*).

## Statistical analysis

In each sample, the following plant diversity indices were calculated (*Zhang, Bai & Han, 2004*), including Patrick index, Pielou index, Shannon–Wiener index, and Simpson index:

$$Pa = S \tag{1}$$

$$D = 1 - \sum (Pi)^2 \tag{2}$$

$$H = -\sum Pi \cdot LnPi \tag{3}$$

$$JP = H/LnS = -\sum Pi \cdot LnPi/LnS \tag{4}$$

$$Pi = (\text{relative abundance+relative plant cover+relative height})/3 \tag{5}$$

*Pa*, Patrick index; *Pi*, Plant species dominance; *D*, Simpson index; *H*, Shannon–Wiener index; *JP*, Pielou index (*Hurlbert, 1971*; *Van der Heijden, Bardgett & Van Straalen, 2008*).

*Fi*, Sample species i/The total number of samples; *S* means the number of species in samples (*Bedford, Walbridge & Aldous, 1999*; *Van der Heijden, Bardgett & Van Straalen, 2008*).

All the statistical analyses were conducted by using SPSS 18.0 software (SPSS Inc., Chicago, IL, USA). The significant difference among different plant diversity and productivity were tested using one-way ANOVA (Analysis of variance). Before one-way ANOVA, we tested the normality and homogeneity of variance assumptions of these plant diversity and productivity which were normally distributed, and thus one-way ANOVA followed by Student's $t$-test was carried out to test the significant difference in plant diversity and productivity at $p<0.05$ level. Further, the dominant driving factors for plant diversity and productivity were performed by CANOCO 5.0 software by RDA (redundancy analysis). Specially, the length of the arrows was determined, and the direction of the arrows for individual driving factors indicated the correlation coefficient among these variables. Plotting was done in Origin 9.2 software. Meanwhile, Pearson correlation, normal regression were used to explore the effect of soil factors on plant diversity and productivity by using R 3.6.0 package (*R Core Team, 2019*). Finally, GAMs (generalized additive models) were used to obtain the explained variation for plant diversity and productivity, which presented in a Venn diagram.

## RESULTS

### Occurrence frequency of all plant species and diversity

The occurrence frequency (*Fi*) values of all plant species were calculated (Table S1). In detail, plant species with the highest frequencies in *Stipa bungeana* communities

mainly included *Euphorbia esula* (0.47), *Artemisia vestita* (0.33) and green bristlegrass (0.33); *Potentilla bifurca,* Artemisia scoparia*, Ixeris chinensis,* and *Salsola ruthenica* were auxiliary species with a lower occurrence frequency and therefore had a lower effect on the community structure. Plant species occurring at higher frequencies in *Agropyron mongolicum* communities mainly included *Ixeris chinensis* (0.67), *Pennisetum flaccidum* (1.00), *Polygala tenuifolia* (0.67), *Lespedeza bicolor* (0.67), and *Heteropappus altaicus* (1.00), and the other plants were auxiliary species with a lower occurrence frequency. Plant species with the highest frequencies in *G. uralensis* communities were *Leymus secalinus* (0.67), *Peganum harmala* (0.47), *Euphorbia humifusa* (0.47), and *C. komarovii* (0.47). In *S. alopecuroides* communities, the occurrence frequencies of plant species were relatively similar except for that of *Pennisetum flaccidum,* and the other plant species were auxiliary species with a lower occurrence frequency. Plant species with the highest frequencies in *Artemisia ordosica* communities were *Agropyron mongolicum* (0.67), *Agropyron mongolicum* (1.00) and *Corispermum hyssopifolium* (0.67), whereas *C. komarovii,* green bristlegrass and *Ixeris chinensis* were auxiliary species with a lower occurrence frequency. The *C. komarovii* communities had high plant species richness, *C. komarovii* and *Oxytropis racemosa* were the dominant species, and species occurring at high frequency were *Polygala tenuifolia, Agropyron mongolicum,* and *Euphorbia esula.*

In Fig. 2, *C. komarovii* and *G. uralensis* communities had the highest number of plant species, followed by *Artemisia ordosica, Stipa bungeana,* and *Sophora alopecuroides,* which all had the same number of plant species. This result indicates that the structural complexity of *C. komarovii* and *G. uralensis* communities was higher than that of other plant communities. *C. komarovii* communities had a higher Patrick index, whereas *Stipa bungeana* communities had a lower Patrick index, and the Patrick index of *C. komarovii* communities was nearly two times that of *Stipa bungeana* communities; there was no significant difference in Patrick index ($p > 0.05$). Simpson's index for the *C. komarovii* and *Sophora alopecuroides* communities was higher than that for the other plant communities, with no significant difference ($p > 0.05$). *Stipa bungeana* communities had a lower Simpson's index value compared with the other plant communities ($p < 0.05$), thus indicating that *Stipa bungeana* communities were relatively homogenous. We found no significant difference in Simpson's index for the *G. uralensis, Agropyron mongolicum,* and *Artemisia ordosica* communities ($p > 0.05$). Pielou's index for the *C. komarovii* and *Sophora alopecuroides* communities was higher than that for the other plant communities, whereas Pielou's index for the *Artemisia ordosica* and *Stipa bungeana* communities was the lowest for these plant communities ($p < 0.05$), suggesting that plant species distributions of *C. komarovii* and *Sophora alopecuroides* communities were uniform.

## Above-ground biomass, Below-ground biomass and R:S ratios

AGB, BGB, R:S ratios of different plant communities showed a higher variation (Table 1). BGB was nearly double than AGB ($p < 0.01$). The proportion of AGB of *Agropyron mongolicum, Stipa bungeana, C. komarovii, G. uralensis, Sophora alopecuroides, Artemisia ordosica* were 40.12%, 38.41%, 18.11%, 17.16%, 30.57%, and 31.26%, respectively. AGB ranged from 43.9 to 300.6 $gm^{-2}$, with a mean value of 132.2 $gm^{-2}$, and BGB ranged from
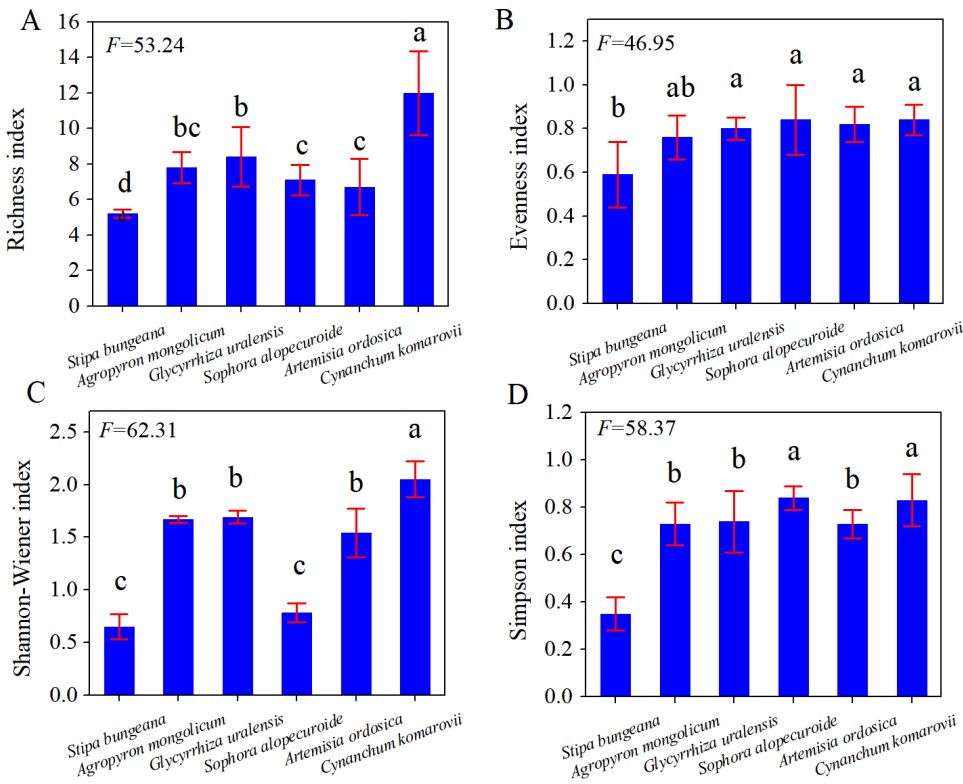

**Figure 2** **Diversity indices of different plant communities in desertified steppe.** Comparison of the richness index (A), evenness index (B), Shannon–Wiener index (C) and Simpson's index (D) for plant communities. Results of each split-plot two-way ANOVA are shown below each corresponding panel. Bars within a panel that share letters have means that differ significantly based on Tukey's test. Error bars represent standard error. $n = 15$ here and subsequently.

80.7 to 682.6 gm$^{-2}$, with a mean value of 290.7 gm$^{-2}$. AGB showed the following order: *Artemisia ordosica* > *Stipa bungeana* > *Agropyron mongolicum* > *Sophora alopecuroides* = *Cynanchum komarovii* > *G. uralensis* with significant differences.

Based on AGB and BGB, we calculated R:S ratios (BGB/AGB) and plotted their frequency distribution. As shown in Fig. 3, R:S ratios had a strong heterogeneity, and ranged between 0.4 and 7.3. Specially, R:S ratios of the *Gramineae* plants (*Stipa bungeana* and *Agropyron mongolicum*) had no significant difference with *Leguminosae* plants (*G. uralensis* and *Sophora alopecuroides*) ($p > 0.05$). Totally, R:S ratios of most plant communities (more than 40%) ranged from 1 to 2. Besides, we found that there was a lower AGB, BGB and R:S ratios compared with the other grasslands around the world (Table 2).

**Relationships among litter, biomass, and the Shannon–Wiener index**

To study the contributions of AGB, BGB, and litter to these plant communities and their growth patterns in the desertified steppe, we evaluated the relationships among litter, biomass, and Shannon–Wiener index using regression models (Fig. 4). There was a positive linear relationship between AGB and BGB, indicating that plant communities

**Table 1  AGB, BGB and R:S ratios of plant communities in desertified steppe.**

| Plant communities | Sample size | AGB/(gm$^{-2}$) | BGB/(gm$^{-2}$) | R:S | | |
|---|---|---|---|---|---|---|
| | | | | Mean | Median | Range |
| *Stipa bungeana* | 15 | 224.9 ± 23.6 b | 335.7 ± 56.9 b[**] | 1.5 c | 1.3 | 0.5–2.6 |
| *Agropyron mongolicum* | 15 | 114.8 ± 15.8 c | 184.1 ± 29.8 d[**] | 1.6 c | 1.1 | 0.7–2.1 |
| *Glycyrrhiza uralensis* | 15 | 43.9 ± 6.9 e | 198.5 ± 23.7 d[**] | 4.5 a | 2.5 | 1.9–6.8 |
| *Sophora alopecuroides* | 15 | 54.4 ± 8.2 d | 262.6 ± 32.1 c[**] | 4.8 a | 3.2 | 2.3–7.3 |
| *Artemisia ordosica* | 15 | 300.6 ± 38.7 a | 682.6 ± 53.2 a[**] | 2.3 b | 1.7 | 1.4–5.1 |
| *Cynanchum komarovii* | 15 | 54.4 ± 9.5 d | 80.7 ± 8.7 e[**] | 1.5 c | 0.9 | 0.4–3.6 |
| Mean | 90 | 132.2 ± 17.1 | 290.7 ± 34.1[**] | 2.7 | – | 0.4–7.3 |

**Notes.**
Values followed by lowercase letters within columns are significantly different at the 0.05 level based on Tukey's method, $n = 15$ here and subsequently.
[**] means AGB and BGB were significantly different ($p < 0.01$).

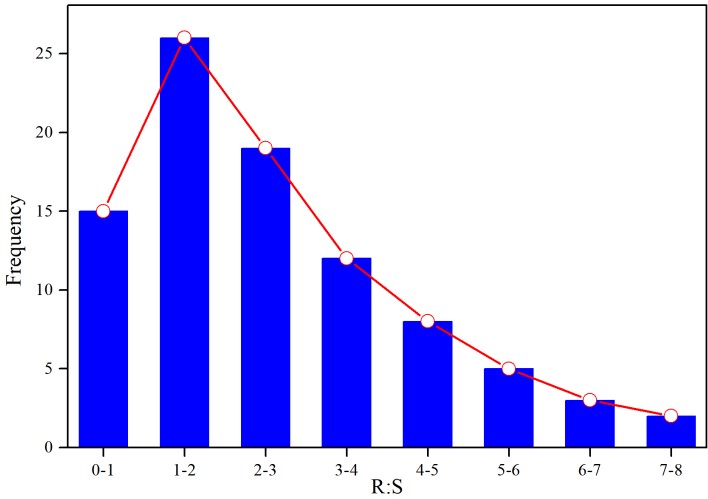

**Figure 3  Frequency distribution of R:S ratios of different plant communities in desertified steppe.**

presented equivalent growth patterns ($R^2 = 0.9025$, $p < 0.001$). Likewise, there was a positive linear relationship between litter and AGB ($R^2 = 0.8485$, $p < 0.001$), BGB ($R^2 = 0.8361$, $p < 0.001$), suggesting that BGB can be accurately estimated by AGB.

Further, we found that the curves between Shannon–Wiener index and AGB were unimodal ($R2 = 0.4572$, $p < 0.05$), whereas no significant relationship between Shannon–Wiener index and BGB and litter ($p > 0.05$). These findings indicated that the degree of resource sharing and the interactions between species of different plant communities changed in the same manner, thus indicating that plant diversity inhibited AGB but did not inhibit BGB. In most grasslands, large amount of studies have confirmed the positive relationship between plant diversity and productivity. In this case, our results also supported this viewpoint. In other words, plant productivity increased with plant diversity because of sufficient natural resources that promoted the reproduction of the plant population at a low level of plant diversity. While at a high level of plant diversity, it was limited by the

**Table 2** Comparison of above-ground biomass (AGB), below-ground biomass (BGB) and R:S ratios of grasslands around the world.

| Country/temperate grassland | | AGB/(gm$^{-2}$) | BGB/(gm$^{-2}$) | R:S | | | Reference |
|---|---|---|---|---|---|---|---|
| | | | | Mean | Median | Range | |
| | Europe | 377.0 | 1903.8 | 3.7 | 3.4 | 1.1–6.9 | *Coupland (1979)* |
| | North America | 207.8 | 1469.6 | 4.4 | 3.7 | 1.2–10.3 | *Coupland (1979)* |
| | Japan | 742.0 | 1415.1 | 4.3 | 4.3 | 1.6–6.9 | *Coupland (1979)* |
| | World | – | – | – | 4.2 | – | *Mokany, Raison & Prokushkin (2006)* |
| China | Meadow steppe | 122.4 | 643.8 | 5.3 | – | – | *Fang, Liu & Xu (1996)* |
| | | 183.4 | 1140.7 | 6.2 | – | – | *Ma & Fang (2006)* |
| | Typical steppe | 135.1 | 553.9 | 4.1 | – | – | *Fang, Liu & Xu (1996)* |
| | | 103.4 | 590.3 | 5.7 | – | – | *Ma & Fang (2006)* |
| | Alpine steppe | 50.1 | 277.7 | 5.5 | – | – | *Yang et al. (2009)* |
| | Desertified steppe Inner Mongolia | 153.6 | 58.01 | – | – | – | *Ma & Fang (2006)* |
| | | 94.3 | 746.3 | 12.7 | 6.8 | 1.2–30 | *Ma & Fang (2006)* |
| | | 182.7 | 2424.1 | 13.0 | 12.6 | 12.5–13.8 | *Ma & Fang (2006)* |
| | | 135.3 | 775.2 | 8.5 | 6.3 | 5.2–6.7 | *Ma & Fang (2006)* |
| | Grassland | 97.0 | 604.2 | – | – | – | *Piao et al. (2009)* |

natural resource, so inter-specific competition intensified, and plant productivity tended to a decrease as a consequence.

## Relationships between soil factors and productivity and diversity

Firstly, we found that soil factors had a large effect on plant diversity and productivity by correlation analysis (Table S2). Specially, plant diversity and soil nutrients presented the positive correlations, whereas plant diversity and soil pH, BD presented the negative correlations, and significant positive correlations were observed among AGB, BGB, litter, and TC ($p < 0.05$).

Secondly, we conducted ordination analysis of plant diversity and productivity; all of plant plots were ordered in NMDS. NMDS showed a higher explanation for plant diversity and productivity on these two axes (first axis/dimension = 52.47, $R^2 = 0.579$, $p < 0.05$; second axis/dimension = 32.09, $R^2 = 0.507$, $p < 0.05$), and plant productivity was strong related to plant diversity (Fig. 5). Further, RDA was used to determine the driving factors affecting plant diversity and productivity (Fig. 6A). In RDA, axes 1 ($p < 0.01$) and 2 explained 62.19% and 19.58% of the data, respectively, suggesting that soil factors (BD, SW, EC, pH, TC, TN, TP, AP, AN, SMB-N, SMB-C) can adequately account for plant diversity and productivity. In addition, the canonical coefficients of soil factors on each axis indicated that soil factors had an obvious effect on plant diversity and productivity (Table S3). Specifically, there was a similar direction for soil factors arrows and AGB and diversity, which indicated a significant correlation (the longer the arrows, the stronger the correlation) of soil factors and plant diversity and productivity. As for AGB and plant diversity, the first axis was correlated with BD, pH, SW, TC, TN, SMB-C, SMB-N, and the second axis was correlated with BD, pH, SMB-C, SMB-N. Totally, these two axes were negatively related to BD and pH. While soil factors had no significant correlation with

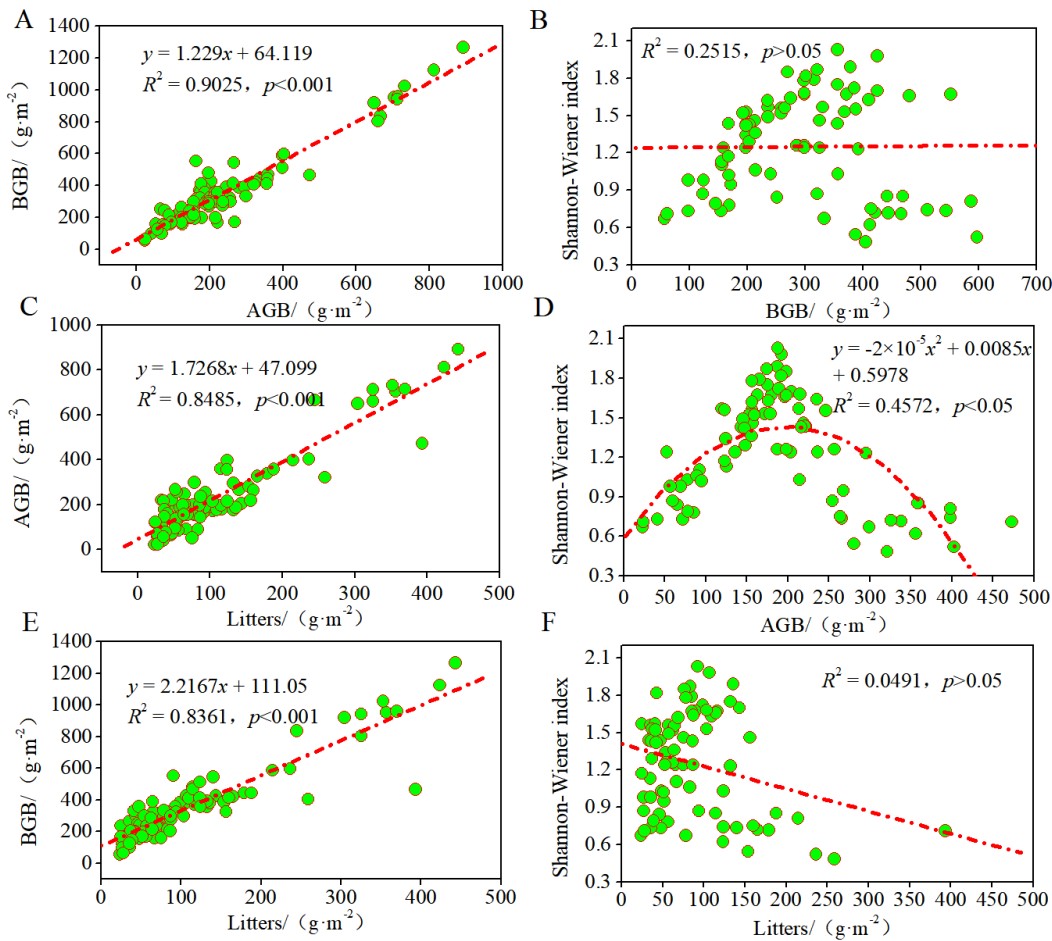

**Figure 4 Relationships among litters, biomass and the Shannon–Wiener index of plant communities.** Regression relationship equations were established ($n = 90$), and the model confidence intervals were 95% and 99% ($p < 0.05$ and $p < 0.1$). (A) regression relationship between AGB and BGB; (B) regression relationship between BGB and Shannon–Wiener index; (C) regression relationship between AGB and litters; (D) regression relationship between Shannon–Wiener index and AGB; (E) regression relationship between BGB and litters; (F) regression relationship between Shannon–Wiener index and litters.

BGB, these findings were in agreement with the results from Pearson correlation analysis, indicating that soil factors had a large influence on plant diversity and productivity. Additionally, TC, TN, SMB-C, and SMB-N can be regarded as the key factors driving plant diversity and productivity in desertified steppe.

Finally, we used generalized additive models (GAMs) to explore the relationship among plant productivity (AGB and BGB), plant diversity, and soil factors (Fig. 6B). GAMs showed that soil factors explained more variation in plant diversity and productivity (78.24%). In detail, soil factors had a greater effect on plant productivity (27.01%) than plant diversity (25.41%).

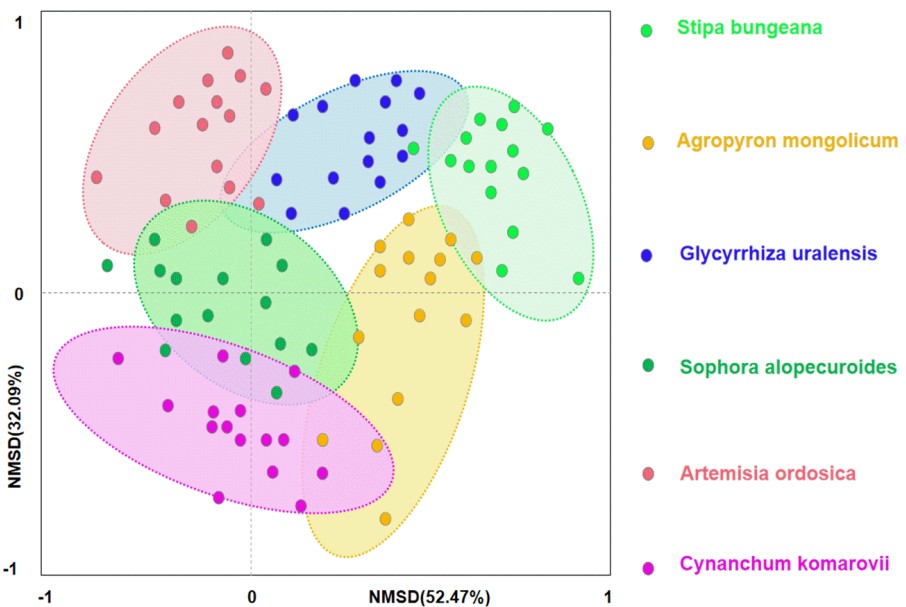

**Figure 5** Nonmetric multidimensional scaling (NMDS) ordination of different plant communities in desertified steppe.

## DISCUSSION

### The dynamics of plant community traits in the desertified steppe

Based on the vegetation and soil investigation, we calculated plant evenness, the dominance index, plant productivity (AGB, BGB), and plant diversity. The same soil texture and characteristics existed among these plant communities, and we found that plant Richness index, Evenness index, Simpson index, Shannon–Wiener index had a large difference among these plant communities (Fig. 2), similar to previous studies (*Sala & Austin, 2000*; *Zhang, Bai & Han, 2004*). Generally, plant community structure was determined by plant species numbers and their composition (*Rejmánek & Richardson, 1996*; *Zobel, 1997*; *Bruelheide et al., 2018*; *Liu et al., 2018*). In our study, the Richness index was similar to Simpson's index. Specifically, the Richness index, Evenness index, Shannon–Wiener index of the *Stipa bungeana* community were lower than the other plant communities, reflecting the plate-block distribution of the *Stipa bungeana* community. However, *G. uralensis* and *C. komarovii* communities had a large plant species number and higher Shannon–Wiener index compared with the *Stipa bungeana* community, supporting the fierce inter-specific competition for natural resources.

AGB, BGB, and R:S ratios of different plant communities showed a higher variation (Table 1), and BGB was higher than AGB. According to optimal allocation, plants in drought ecosystems tend to accumulate more BGB for more nutrient absorption to adapt to the fragile environment (extreme drought). Thus, our findings are agreement with previous research from grasslands in arid region around the world (*Hedlund, Santa Regina & Van der Putten, 2003*; *Tilman, Reich & Knops, 2006*; *Cardinale, Wright & Cadotte, 2007*;
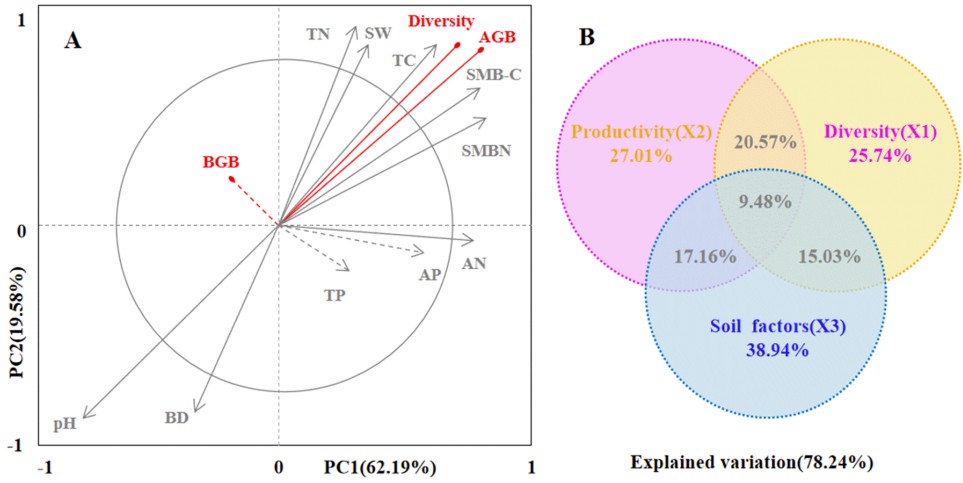

**Figure 6** **The driving factors of plant diversity and productivity.** Redundancy analysis (RDA) showing the relationships between soil factors and productivity and diversity (A). In the biplot, productivity and diversity are highlighted in red, and soil factors are in gray. Each arrow represents the eigenvector corresponding to an individual variable. PC1 accounted for 62.19% of the overall variance, and PC2 accounted for 19.58% of the overall variance. The results of the generalized additive models (GAMs) for plant productivity and diversity (B). The GAMs analyses led to the following fractions: pure effect of productivity; pure effect of diversity; pure effect of soil factors; joint effects of productivity, diversity and soil factors; and ex plained variation.Soil factors (SW-soil water content, BD-bulk dens ity, EC-electrical conductivity, pH-pH value, TC- total carbon , TN-total nitrogen, TP-total phosphorus, AP-available phosphorus, AN-available nitrogen, SMB-C-soil microbial biomass carbon, SMB-N-soil microbial biomass nitrogen) are represented as arrows, and the strength of their impacts is directly proportional to the length of the arrow lines.

*Chen et al., 2018*). However, R:S ratios of most plant communities (more than 40%) ranged between 1 and 2, which were lower than the other grasslands around the world (Table 2). Generally, the lower R:S ratios indicated that these plant communities presented high plasticity for their own survival and reproduction. In order to adapt to the fragile conditions, plant vegetative organs need to more biomass in their growth, which is called to the survival strategy in the desertified steppe (*Whittaker & Heegaard, 2003*; *Cardinale, Wright & Cadotte, 2007*). Thus, R:S ratios were lower compared with the other grasslands around the world. Besides, there were some reasons resulting in these discrepancies: (1) There are some special differences in extremely environmental conditions (e.g., climate and soil factors), and also the plant community composition exhibited large differences. For example, the size of plant species in this region was small due to the arid environment (wind erosion, water loss and soil erosion, extreme drought, and so on), leading to the lower AGB and BGB. However, in order to adapt to the extremely environmental conditions, these plant species must enhance the growth of their roots to absorb soil nutrients, thus resulting in the higher BGB compared with AGB. (2) Over the history of our study area, long-term grazing, the fragile environment, the low levels of precipitation, and the interference of human activities have caused the lower AGB and BGB compared with the global level (*Tang et al., 2018*).

## The linkage and driving factors of plant diversity and productivity in the desertified steppe

To explore the contributions of AGB, BGB and litter to plant diversity and productivity, we tested the relationships between AGB, BGB, litter, and plant diversity by using a regression model (Fig. 4). We found a positive linear relationship between AGB and BGB, which presented an equivalent growth pattern ($R^2 = 0.9025$, $p < 0.001$), and this result agrees with most studies from the other grasslands in the world (*Hedlund, Santa Regina & Van der Putten, 2003*; *Ma & Fang, 2006*; *Cardinale, Wright & Cadotte, 2007*; *Bai, Wu & Clark, 2012*). Further, the positive linear relationships between litter and AGB ($R^2 = 0.8485$), BGB ($R^2 = 0.8361$) indicated that AGB and BGB were mainly dependent on litters. In recent decades, the dominant view supported the hump-shaped or unimodal pattern between plant diversity and productivity around the world (*Fang, Liu & Xu, 1996*; *Mittelbach, Steiner & Scheiner, 2001*; *Tilman, Reich & Knops, 2006*; *Yang et al., 2018*). For example, Tilman et al. build a specific model to predict the number of plant species that coexisted on a limited resource base, and showed that plant diversity first increased and then decreased when the supply of natural resources was limited (*Liu et al., 2018*; *Tang et al., 2018*). In our study, the curves between plant diversity and AGB were unimodal ($R^2 = 0.4572$, $p < 0.05$), supporting the previous viewpoint (*Mittelbach, Steiner & Scheiner, 2001*; *Tilman, Reich & Knops, 2006*; *Bai, Wu & Clark, 2012*). The hump-shaped form indicated that plant productivity increased at a low level of diversity but decreased at high level of diversity (*Bai, Wu & Clark, 2012*; *Tang et al., 2018*). At a low level of plant diversity, plant productivity increased with diversity because the sufficient natural resource promoted the reproduction of the plant population (*Mittelbach, Steiner & Scheiner, 2001*; *Hedlund, Santa Regina & Van der Putten, 2003*; *Ma & Fang, 2006*). At a high level of diversity, it was limited by the natural resource, so inter-specific competition was intensified, and plant productivity tended to decrease as a result (*Mittelbach, Steiner & Scheiner, 2001*). However, there was no significant relationship between plant diversity and BGB. The reason was that AGB was strong related to plant diversity; when AGB got to the peak, plant diversity decreased because of the fierce competition among plant species, but the fierce competition did not affect BGB since BGB was mainly dependent on soil nutrients below-ground, thus plant diversity had no effect on BGB ($p > 0.05$), so the findings are agreement with those results reported from Leibold (*Leibold, Holyoak & Mouquet, 2004*) and Tilman (*Tilman, Reich & Knops, 2006*).

Additionally, we tested the relationships between plant diversity, productivity and soil factors in the desertified steppe using Pearson correlation coefficients (Table S2). We found that soil factors strongly affected plant diversity and productivity, which were similar to the results from other studies (*Liu, Zhao & Zhao, 2012*; *Bai, Wu & Clark, 2012*). Actually, plant diversity and soil nutrients (TC, TN, TP, AN, AP, SMB-C, and SMB-N) presented the positive correlations, whereas plant diversity and soil pH, BD presented the negative correlations, suggesting that soil nutrients positively contributed to plant diversity, whereas soil pH and BD had a negative contribution to plant diversity. In addition, GAMs showed that soil factors explained more variation in plant diversity and productivity (78.24%). In detail, soil factors had a greater effect on plant productivity (27.01%) than plant diversity

(25.41%). RDA results demonstrated that soil factors accounted for plant diversity and productivity. Specifically, TC, TN, SMB-C, and SMB-N can be regarded as the key factors driving plant diversity and productivity in desertified steppes in northwestern China.

## CONCLUSIONS

In summary, we found a positive relationships between plant productivity and diversity in a desertified steppe, northwestern China. Specifically, the linear relationship between AGB and BGB, showed that plant communities presented an equivalent growth pattern. Further, the curves between plant diversity and AGB were unimodal, indicating that plant productivity increased at a low level of diversity but decreased at a high level of diversity, whereas there was no significant relationship between plant diversity and BGB. In addition, RDA indicated that soil factors had a strong effect on AGB, while soil factors had no significant correlation with BGB. GAMs showed that soil factors explained more variation in plant diversity and productivity. TC, TN, SMB-C, and SMB-N can be regarded as the key factors driving plant diversity and productivity in desertified steppes in northwestern China. Notably, the relationships between productivity and diversity are driven by other climatic conditions. Therefore, we should further concentrate on the other climatic conditions to explain the relationships between plant diversity and productivity and their driving factors in this region.

## ACKNOWLEDGEMENTS

We would like to acknowledge the great help from Shaoshan An in State Key Laboratory of Soil Erosion and Dryland Farming on the Loess Plateau, Northwest A&F University, and we thank LetPub/Accdon for polish the language.

### Funding

This work was supported by the National Natural Science Foundation of China (31660168), and the First Class Discipline Construction Project in Colleges and Universities of Ningxia Hui Autonomous Region (Ecology, No. NXYLXK2017B06). The funders had no role in study design, data collection and analysis, decision to publish, or preparation of the manuscript.

### Grant Disclosures

The following grant information was disclosed by the authors:
National Natural Science Foundation of China: 31660168.
the First Class Discipline Construction Project in Colleges and Universities of Ningxia Hui Autonomous Region: Ecology, No. NXYLXK2017B06.

### Competing Interests

The authors declare there are no competing interests.

### Author Contributions

- Yang Yang and Bingru Liu conceived and designed the experiments, performed the experiments, analyzed the data, contributed reagents/materials/analysis tools, prepared figures and/or tables, authored or reviewed drafts of the paper, approved the final draft.

### Field Study Permissions

The following information was supplied relating to field study approvals (i.e., approving body and any reference numbers):

Field experiments were approved by The Grassland Committee of Ningxia Hui Autonomous Region (#15032).

### Data Availability

Raw data are available in Tables S1–S3.

### Supplemental Information

Supplemental information for this article can be found online at http://dx.doi.org/10.7717/peerj.7239#supplemental-information.

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
