# Peer review of "Testing relationship between plant productivity and diversity in a desertified steppe in Northwest China"

_PeerJ, doi:10.7717/peerj.7239_

## Round 0.1 · original submission · Major Revisions

The reviewers found the manuscript to be if interest but commented on multiple technical faults, being overall poorly written, and containing unnecessary repeats as well as typos. Please address the point-by-point suggestions of the three reviewers. In addition, a professional English language editor following a restructure of the paper is needed.

Reviewer 1 ·

Basic reporting

Study seems interesting and data are well publishable, but manuscript need minor correction. According to me Manuscript fulfill journal standard.

The manuscript contains a multitude of typographical-, comma- and grammatical errors, sentence fragments and mixed use of capital and small initial letters. Most of these errors could have been easily avoided by using the spell check function found in commercial word-processing software.

Plagiarism is 35% I would suggest author to bring it down to 20%. Plagiarism report is attached.

Below are the Mistakes I found


51 should be “in the form of” insted “in form of”
29 should be “role in the controling” insted “roles in the control”
43 should be “the dominant” instead of “dominant”
58 should be “plant” instead “plant plant”
63 should be “plant” instead “plant plant”
64 should be “plant” instead “plant plant”
65 should be “has” instead of “have”
91 should be “China” insted “China.”
96 should be “the desertified” insted “desertified”
131 should be “aluminium” insted “aluminum”
133 should be “the soil and levelled” insted “soil and leveled”
229 should be “These findings” insted “This findings”
224 should be “which” insted “whcih”
247 should be “strong” insted “strongly”
297 should be “pattern” insted “patterns”
314 should be “because of” insted “because”
315 should be “led” Instead “leaded”
317 should be “decreases” insted “decreased”
343 should be “similar” instead “similarly”
354 should be “ were dependent on the soil total’ instead “was dependent on soil total”
359 should be “the soil” insted “soil ”

Experimental design

Experimental Design is ok.

Validity of the findings

Manuscript fulfill the AIM of the study.
Numerous studies have reported diversity patterns along natural productivity gradients. So there is no novelity of work.

Annotated reviews are not available for download in order to protect the identity of reviewers who chose to remain anonymous.

Reviewer 2 ·

Basic reporting

I find there are too many grammar mistakes in the text as follows, and substantial professional language editing should be used to improve the English used throughout the article.
L13 “different grassland” should be “different grasslands”.
L30 “played an important roles” should be “played important roles”
L37 “an ecological system” should be “an ecosystem”
L70 “most research” should be “most researches”
L103 “mean temperatures is” should be “mean temperatures are”
……

Experimental design

The experimental design description is rather vague. Six dominant plant species chosen in this study were selected. However, the authors did not mention what is the distance between the six plant species? Are they in the same community? They should clarified the experimental design.

Validity of the findings

1. The results section needs to be more explicit, and grammatical errors need further modification as follows.
L219-220 What is the meaning of this comparison? If so, please move it the Discussion part.
L223-224 “by using model” should be “using model”.
L249-250 should be moved to the MMs part.
L254-258 is a very long sentence. It should be changed to several short sentences and clarified the meaning.
……
2. The conclusion is only a repetition of the results and is not highly summarized.

Additional comments

The authors tried to test the association of plant community productivity and diversity. Although I appreciate the effort of the work in the natural ecosystems, the conclusion presented in the manuscript is less surprising or novel.
1. The MS is not overall well written, and contains some unnecessary repeat and grammar mistakes. A professional English language editor following a restructure of the paper is needed.
2. The authors also analyzed the relationships between soil physi-chemical properties and plant diversity and productivity. I advise the authors should focus on the effects of soil physi-chemical properties on the relationships of plant diversity and productivity, rather than on plant diversity or productivity.
3. L93-94 What is the difference between Q1 and Q2? The authors should clarify the scientific questions.
4. L185-186 should be moved to the Discussion part.
5. L231-232 What is the internal mechanism of “that plant diversity inhibited AGB but did not affect BGB”?
6. The Discussion part have too many repetitions with the results section, and there is no further explanation for the reasons of the results.
7. L286-292 These three reasons should be clarified.

·

Basic reporting

Literature references, sufficient field background provided
Self-contained with relevant results to hypotheses

Experimental design

Original primary research within Aims and Scope of the journal.
Research question well defined, relevant & meaningful. It is stated how research fills an identified knowledge gap.
Rigorous investigation performed to a high technical & ethical standard.
Methods described with sufficient detail & information to replicate.

Validity of the findings

Data is robust, statistically sound, & controlled.

Conclusions are well stated, linked to original research question

Additional comments

The relationships between vegetation production and species diversity is a core question in ecosystem function. There are many different patterns were reported over global grassland, but lack of the related research in the desertified steppe. Thus, it is a valuable research, but there have some issues should be corrected moderately before publication.

ABSTRACT , I suggested that the author should re-write the it, the clear expression is needed
e.g. line 13-18, line 23-28.
Line 20-21, plant diversity inhibited AGB but did not inhibit BGB, why? What is the mechanism ?
Line 29-30 Finally, the interactions among soil nutrients and their influence played an important roles in the control of plant diversity and productivity in desertified steppe, northwest China.
I hold that this is the general phenomenon, we can conclude it without some research. Consequently, I hope the author can refine the important findings.


Introduction, Additionally, studies of the relationship between productivity and diversity in grasslands have been conducted primarily in Europe, Africa, and North America, and there are many potential problems associated with the data, including sample sizes and the surrogates used for productivity (Van et al., 2001; Bai et al., 2012; Cardinale et al., 2015; Trax et al., 2015). Understanding the interactions among resource availability, diversity and productivity is relevant for the preservation, management and restoration of native communities and may be key in successfully restoring these species-rich ecosystems (Tilman et al., 2012; Cardinale et al., 2015; Trax et al., 2015; Craven et al., 2016).
Line 82-88, How to link this sentence-there are many potential problems associated with the data, including sample sizes and the surrogates used for productivity with next sentence? Please revise it.
In another word, what is the relationships between sample size….. and your topic?

Methods,
Line 115-117, these values approximate the above-ground net primary productivity (ANPP) in temperate grasslands. These values contained the AGB and BGB?

Line 114 2.2 Experimental design might be changed into 2.2 sampling methods
Line 136 2.3 Methods might be changed into 2.3 Soil sample analysis?

Result
3.1 Occurrence frequency of all plant species and diversity indices
3.5 Relationships between soil factors and plant species on the basis of redundancy analysis (RDA)
Please re-write these sections, I hold that too long to express the most important result clearly.

4 Discussion
4.1 Species occurrence and diversity of different plant communities
4.2 AGB, BGB and R:S ratios of different plant communities
4.3 Relationship among litters, biomass and Shannon-Wiener index of different plant communities
4.4 Testing associations of diversity-productivity of plant communities

I suggested that authors should divide the DISCUSSION into two section;
4.1 The dynamics of community traits
4.2 The linkage of diversity-productivity in the desertified steppe

References
Grazing enhances soil nutrient effects on trade-off between above and belowground biomass in alpine grasslands of the Tibetan plateau
Meta-analysis of relationships between the environmental factors and the aboveground biomass in alpine grassland, Tibetan Plateau
Some reports are about the relationships between vegetation biomass with soil properties, you can understand it.

Figures, tables and data are enough for this research.

Sorry, I am not the native English speaker, so ,I can not give you some grammar comments.

---

## Round 0.2 · Minor Revisions

The three reviewers agreed that a significant efforts were undertaken in order to improve the manuscript.

There are several small issues that I encourage you to correct before re-submission. One of the reviewers identified there was 25% plagiarism detected in the text by the Reviewer 1. I have examined the attached plagiarism report and the problem does not appear to be that severe. Frequently, plagiarism software is overly sensitive, and it can label parts containing legitimate terms (e.g. Next Generation Sequencing) as being plagiarized.
It is recommended to re-phrase these parts, if possible, and add/or citations to the source material, if external sources were used.

I also suggest to streamline the abstract, leaving only the most important results and conclusions there.

In the Statistical Analysis section, would it be possible to typeset the mathematical formulas using better equation editor? In the current format the formulas are hard to read. In the same section you mention conducting ANOVA test. Were ANOVA assumptions satisfied?
Line 210 has a heading "AGB, BGB, and R:S ratios of different plant communities". I suggest not using abbreviations in section headings.

Since it is important to improve the presentation of your manuscript, it would be beneficial at this stage to involve a technical editor to fix grammatical problems.

Reviewer 1 ·

Basic reporting

The study was well planned and executed. The paper is acceptable, but I have some concerns.

Abstract need to be rewritten, concluding section is missing and it looks more of introduction.

The manuscript contains some multitude of typographical-, comma- and grammatical errors, sentence fragments and mixed use of capital and small initial letters, I have corrected them see attached file

Although the plagiarism is 25% but some sentence needs to be a paraphrase. (see the attached plagiarism report)

I suggest accepting the manuscript after the above changes are made by the author.

Experimental design

Perfect

Validity of the findings

Manuscript fulfill the AIM of the study.

Annotated reviews are not available for download in order to protect the identity of reviewers who chose to remain anonymous.

Reviewer 2 ·

Basic reporting

There are still some grammatical problems, and the authors should carefully improve the language before further processing.

Experimental design

No comment

Validity of the findings

No comment

Additional comments

I am appreciate with the correction and clarification on the comments that previous raised. However, I find there are still some grammatical problems, and the authors should carefully improve the language before further processing.

·

Basic reporting

The revised MS is fine. The expression is smooth, reference is enough for supporting the field background. Besides, the structure has been corrected according to my comments.

Experimental design

The MS is fit for the aim and scope of Peer J

Validity of the findings

Data is robust, and conclusion is well proofed.

Additional comments

The MS has been revised well, thus, I recommend publishing this paper.

---

## Round 0.3 · Minor Revisions

Thank you for incorporating the suggestions. However, I encourage you to do two things before submission of the final version:

1) typset equations using MS Word Equations Editor. Formulas are hard to read
2) Explain in the text how the ANOVA assumptions were satisfied.

Adequately addressing these two items should be sufficient to satisfy the acceptance criteria of PeerJ

---

## Round 0.4 · accepted · Accept

Thank you very much for typesetting the equations and explaining how the ANOVA assumptions are satisfied in your experiment. This was the final requirement to make your paper suitable for publication.